# The Use of Probiotics Combined with Exercise Affects Thiol/Disulfide Homeostasis, an Oxidative Stress Parameter

**DOI:** 10.3390/nu14173555

**Published:** 2022-08-29

**Authors:** Yıldırım Kayacan, Aybike Zeynep Kola, Stefano Guandalini, Hayrullah Yazar, Mehtap Ünlü Söğüt

**Affiliations:** 1Faculty of Yasar Dogu Sports Sciences, Ondokuz Mayıs University, Samsun 55139, Turkey; 2Pediatrics-Gastroenterology, University of Chicago Medicine, Chicago, IL 60637, USA; 3Department of Medical Biochemistry, Sakarya University Faculty of Medicine, Sakarya 54050, Turkey; 4Faculty of Health Sciences Department of Nutrition and Dietetics, Ondokuz Mayıs University, Samsun 55139, Turkey

**Keywords:** thiol, disulfide, oxidative stress, probiotics, exercise

## Abstract

Background: Intestinal microbiota play a role in the health and performance of athletes, and can be influenced by probiotics. Thus, in this study, we aimed to investigate the effect of the use of probiotics combined with chronic exercise on the thiol/disulfide homeostasis, a novel marker of oxidative stress. Methods: Male Wistar rats were randomly divided into four groups: control (Cn), exercise (Ex), probiotics (P), and probiotics + exercise (PEx). A capsule containing 6 × 10^8^ CFU of *L. rhamnosus*, *L. paracasei, L. acidophilus,* and *B. lactis* was given daily for eight weeks to all the experimental animals. The total thiol (TT, μmol/L) and native thiol (NT, μmol/L) concentrations were measured to determine the oxidative stress parameters. The dynamic disulfide (DD, %), reduced thiol (RT, %), oxidized thiol (OT, %), and thiol oxidation reduction (TOR, %) ratios were analyzed. Results: The TT level was found to be significantly higher in the Ex group (p = 0.047, η^2^ = 0.259). The DD level, a marker of oxidation, was significantly lower in the PEx group (p = 0.042, η^2^ = 0.266); the highest value of this parameter was found in the Ex group. The use of probiotics alone had no effect on thiol/disulfide homeostasis. Conclusions: We showed, for the first time, that probiotics administered “with exercise” decreased dynamic disulfide and significantly reduced oxidative damage. Therefore, we speculate that the use of probiotics in sports involving intense exercise might be beneficial to reduce oxidative stress.

## 1. Introduction

Intestinal microbiota are very important to the health and performance of athletes. Today, there is evidence-based information that indicates intestinal microbiota are regulated by various environmental conditions (nutrition, stress, mode of delivery, etc.), including exercise [1]. Intestinal microbiota are composed of about 10^14^ microorganisms and various disorders are known to affect the intestinal microorganism balance [2]. Probiotics are recommended to prevent or reverse such changes.

Probiotics are live microorganisms that have a positive effect on a host when given in sufficient amounts [3]. It has been reported that *Lactobacillus* group bacteria possess antioxidant effects; *Lactobacillus fermentum* strains, which have antioxidant properties, in fact, cause the release of manganese superoxide dismutase (Mn-SOD) which is an important defense tool in the prevention of lipid peroxidation, and contain significant levels of glutathione [4].

Microbiota may also positively affect exercise performance. In fact, it has been reported that, in athletes, healthy microbiota positively affect critically important metabolic parameters such as energy metabolism, oxidative stress, and hydration status [5]. Additionally, it has been reported that healthy microbiota have curative effects on the immune system suppression that athletes often experience due to strenuous and long-term exercise [6]. In recent years, new studies have been conducted on the importance of gut microbiota in controlling oxidative stress and inflammatory response in athletes, and their key roles in exercise metabolism and energy use [7,8].

Thiol groups are organic, essential, and powerful antioxidant molecules that contain the sulfhydryl (-SH) group which defends organisms against the destructive effects of oxidative stress. High-intensity exercises induce high oxygen consumption, especially in skeletal muscles. With an increase in metabolic activity during exercise, oxygen activity also increases. As a result, reactive oxygen species (ROS) are released. Long-term, high-intensity exercises lead to an excessive increase in blood and skeletal muscle oxidative stress, resulting in the breakdown of macromolecules [9,10]. Thiols can enter oxidation reactions via oxidants and cause disulfide formation. When oxidative stress is increased, oxidation of cysteine residues can lead to the formation of disulfides. However, this process is always reversible. Disulfide bonds can also be reduced in thiol groups, thus, resulting in thiol/disulfide homeostasis. Dynamic thiol/disulfide homeostasis plays important roles in cell signaling mechanisms, transcription factors, enzymatic regulation, activation, apoptosis and signal transduction, antioxidant protection, and detoxification [11].

In our previous studies, we determined the effects of exercise and supplement use on the thiol-disulfide balance [10,12]. We hypothesized that, in order to reduce the oxidative damage induced by exercise, the use of probiotics might beneficially affect the microbiota, and thus, control inflammation and redox levels in rats that exercise. In this context, for this study, we formed the following hypotheses: (1) Chronic intense exercise affects redox metabolism. (2) Probiotics are effective in reducing oxidative stress. (3) The use of probiotics combined with exercise positively affects the thiol-disulfide balance. This study was designed with the aim of examining these hypotheses and analyzing the emerging relationships.

## 2. Materials and Methods

### 2.1. Animals

The experimental protocol was approved by the Animal Ethics Committee (no. 2016/36). A total of 32 male albino Wistar rats, weighing from 180 to 200 g and 12 weeks old, were kept under standard environmental conditions for temperature (21.5 °C) and humidity (60 § 1%) and on a 12:12 h light/dark cycle. The rats were kept in a well-ventilated room and allowed free access to a standard pellet diet along with water ad libitum. The rats were randomly divided into four groups: control (Cn), exercise (Ex), probiotics (P), and probiotics + exercise (PEx) groups. The research protocol is schematically reported in Figure 1.

### 2.2. Probiotic Administration Protocol

A pool of probiotics that included *L. rhamnosus*, *L. paracasei*, *L. acidophilus,* and *B. lactis* (Solgar, NJ, USA) were given for eight weeks daily (6 × 108 CFU of each strain, final concentration of 1.8 × 109 CFU of bacteria). Prior to gavage, the probiotics were diluted in 1 mL of sterile water [13]. This product was chosen because it contains important and major phyla of intestinal microbiota, it possesses a high number of microorganisms per tablet, and it is standardized.

### 2.3. Exercise Protocol

Exercise and the administration of probiotics were performed 5 day/week for 8 weeks. The Conformite Europeene (CE)-certified four-lane animal treadmill (May Time 0804, Animal Treadmill) with adjustable settings for rate, distance, running time, speed, and inclination, and a built-in memory to store data was used for the exercise experiments. To avoid any stress that may have arisen during physical exercise, all rats were preliminarily subjected to a conditioning exercise series at the lowest speed in 5-min sessions for 10 d. After the treadmill adaptation period, the control group rats were put in cages with the standard conditions until surgery, whereas the exercised groups continued to be trained according to the treadmill exercise protocol. The exercise workload consisted of running at a speed of 2 m/min for the first 5 min, 5 m/min for the next 5 min, and then 8 m/min for the last 20 min, with a 0° angle inclination.

### 2.4. Biochemical Parameters

At the end of the experiment, the rats in all groups were starved overnight for 12 h, and sacrificed under ketamine hydrochloride (10 mg/kg intraperitoneally) anesthesia. Blood from the heart was collected after entering the abdominal and thoracic cavities, into biochemistry tubes using 10 mL syringes (5 mL). The samples were centrifuged (1500 g for 10 min) after waiting 30 min. Subsequently, the separated serums were stored in tubes with Eppendorf (Isolab centrifuge tubes 2.0 mL, flat cap, without skirt) with −80° cap. The samples were transferred to the laboratory with a dry ice system at 24 h before the working day. The incoming samples were again microcentrifuged, and the test parameters were studied in Rel Assay Diagnostics kits. The biochemical analysis of this study was also performed at the Research Hospital Clinical Biochemistry Laboratory (Beckman Coulter, fully automated chemistry analyzer AU 680, serial number 2016024580, made in Japan). In this study, dynamic thiol/disulfide homeostasis in the serum samples of rats was identified by using an automated method developed by Erel and Neselioglu [14].

### 2.5. Thiol/Disulfide Analyses

#### 2.5.1. Precision

Three levels of plasma pools were tested to determine the precision of the new test. The plasma pool that had high disulfide levels was obtained from the samples of patients with diabetes mellitus. The plasma pool with medium disulfide levels was obtained from the samples of healthy persons. The plasma pool with low disulfide levels was obtained from the samples of patients with urinary bladder cancer. The percentage coefficient variation (%CV) was 4 ($\bar{X}$ = 29.12 and σX = 1.2) for high levels, 5 ($\bar{X}$ = 16.03 and σX = 0.79) for medium levels, and 13 ($\bar{X}$ = 7.15 and σX = 0.98) for low levels.

Total thiol (–SH + –S–S–) and native thiol (–SH) concentrations in the samples were measured by using Ellmann’s and modified Ellmann’s reagent. The native thiol content was subtracted from the total thiol content, and half of this difference gave the amount of dynamic disulfide bonds (–S–S–). In addition, the (–S–S–) × 100/(–SH), (–S–S–) × 100/(–SH + –S–S–), and –SH × 100/(–SH + –S–S–) ratios were calculated using these parameters.

#### 2.5.2. Analytical Recovery

The percentage of recovery of the novel method was determined via the addition of 200 μM oxidized glutathione to plasma samples. The mean percent recovery was 98–101%.

#### 2.5.3. Linearity

The linearity of the native thiol measurement was the same as that of the Ellman’s reagent assay. Serial dilutions of the glutathione solution were generated. The upper limit of the linearity for the native thiol measurement was 4000 μM. Linearity of the total thiol measurement was also dependent on the amounts of NaBH4 and formaldehyde concentrations. Serial dilutions of the oxidized glutathione solution were also generated. The upper limit of the linearity for the disulfide measurement was 2000 μM. Dilution of plasma samples did not affect the novel assay.

#### 2.5.4. Lower Detection Limit

The detection limit of the assay was determined by evaluating the zero calibrator 10 times. The detection limit, defined as the mean value of zero calibrator + 3 standard deviations (SD), was 2.8 μM.

#### 2.5.5. Analytical Sensitivity

As the slope of the calibration line, analytical sensitivity was found to be 7.9 × 10^−4^ absorbance/amount, (A × (μM) ^−1^).

#### 2.5.6. Interference

It was found that hemoglobin, EDTA, citrate, and oxalate did not interfere with the assay developed, but bilirubin did negatively interfere with the assay. Lipaemic and uraemic plasma samples did not interfere with the assay. The plasma and serum samples could be used as samples.

#### 2.5.7. Storage

Storage at 4 °C for 1 day led to a 7% decrease in the native thiol amount and 170% increase in the disulfide amount (total thiol, native thiol, and disulfide levels of fresh and stored plasma samples were 391 μmol/L, 357 μmol/L, 17 μmol/L (fresh) and 391 μmol/L, 333 μmol/L, 29 μmol/L (stored), respectively). Plasma native thiol, total thiol, and disulfide concentrations were not affected by storage at −80 °C for 3 months.

### 2.6. Statistical Analyses

All results are presented as the means ± standard error of the mean (SEM). The statistical analyses were performed using SPSS ver. 21.0 (SPSS Inc., Chicago, IL, USA). The normality of the data was tested before the analyses. One-way analysis of variance (ANOVA) and Tukey HSD post hoc tests were performed for normal distribution data. The Eta squared (η^2^) formula was used to calculate the effect size. (η^2^ = SS between/SS total). For all statistical tests, *p* < 0.05 was considered to be statistically significant.

## 3. Results

There were no statistical differences found among the groups in NT (η^2^ = 0.228), RT (η^2^ = 0.129), OT (η^2^ = 0.187), and TOR (η^2^ = 0.074) levels. However, it was determined that treadmill exercise (Ex) significantly increased the total thiol (TT) values in rats as compared with the PEx group (η^2^ = 0.259, *p* = 0.047, PEx 490.14 ± 17.87 vs. Ex 569.29 ± 19.753 μmol/L, Table 1 and Figure 2). The highest rate of oxidized thiol was observed in the PEx group (TOR 19.29 ± 0.286%). However, despite this finding, the dynamic disulfide value, which has been accepted as an important indicator of oxidation, was also detected in the lowest PEx group (η^2^ = 0.266, *p* = 0.042, 706.29 ± 28.052%). This value was highest in the Ex group (829.43 ± 32.162%, Figure 3). Descriptive statistical findings and significance levels of the groups are given in Table 1.

Descriptive statistical findings and significance levels of native thiol (NT), total thiol (TT), dynamic disulfide (DD), RT, oxide thiol (OT), and TOR values of the groups are indicated. The highest serum DD, OT, NT, and TT values were observed in the Ex group. These parameters were detected at the lowest level in the PEx group.

## 4. Discussion

In this study, the effect of probiotic supplementation on thiol/disulfide homeostasis was investigated in rats that were subjected to chronic treadmill exercise for 8 weeks. Although the effect of probiotics on oxidative stress has been investigated in studies in humans and animals [15,16], no other study was found that investigated the effect of multi-strain probiotics (1.25 billion × 4 varieties, 5 billion live microorganisms in total) on oxidative stress caused by chronic exercise. We believe that this study will contribute to research on the use of probiotics in the redox mechanism.

### 4.1. The Use of Probiotics Alone Did Not Affect Oxidative Stress

In our study, the use of probiotics alone did not have a significant effect on oxidative stress. Nevertheless, probiotics have been clinically proven to be beneficial in chronic diseases such as gastrointestinal disorders, autoimmune diseases, obesity, insulin resistance, type 2 diabetes, and non-alcoholic fatty liver disease [17,18,19]. These effects are achieved through alternative pathways, possibly including the displacement of other microorganisms due to a competitive environment created by probiotics [20]. In addition, it has been reported that probiotics function as protective agents against oxidative stress such as lactoperoxidase, suppressing bacterial infections by reducing intestinal pH, producing some digestive enzymes and vitamins, and producing antibacterial substances such as hydrogen peroxide, diacetyl, acetaldehyde, and lactoperoxidase [21]. The findings in different studies have shown that the use of probiotics improves antioxidant capacity in humans and experimental animals [21,22]. However, in this study, it was determined that the use of probiotics alone did not have a positive effect on oxidative stress. It has been reported in the literature that factors such as the strain used, duration of use, gender, and disease level play roles in the effect of probiotics on oxidative stress [23,24]. Thus, in spite of a commonly held opinion that probiotics exert positive effects on antioxidant levels, we think that their effects detected on thiol/disulfide homeostasis may well be due to the other factors, as mentioned above.

### 4.2. The Use of Probiotics Combined with Exercise Reduced Oxidative Stress

In the present study, we showed that chronic exercise combined with the use of probiotics significantly reduced oxidative stress. Our findings, therefore, open the way to further research on the role of probiotics in ameliorating athletes’ response profiles to oxidative stress. Although it has been stated that athletes will be more exposed to the risk of exercise-induced free radical flow, especially with increased intensity and duration [25,26,27], different studies have shown that people who exercise regularly and athletes adapt to such a program over time and are more resistant to oxidative damage [28,29]. As a general principle, antioxidant enzyme activity increases significantly in humans and in rats who exercise regularly. An increase in antioxidant activity also responds by preventing lipid peroxidation caused by increased oxidative stress induced by exercise [30]. In addition, individuals who exercise regularly have an additional advantage over sedentary people. In fact, training supports the development of the activity of many major antioxidant enzymes and the general antioxidant level [31]. An increase in antioxidant activity also responds by preventing lipid peroxidation, a marker of exercise-induced oxidative stress [30]. Several published investigations support the findings of our study. For example, Martarelli et al. investigated the effects of *Lactobacillus rhamnosus* IMC 501 and *Lactobacillus paracasei* IMC 502 probiotic strains on oxidative stress in athletes during 4 weeks of intense training (~10^9^ cfu per day). As a result of the study, it was determined that these probiotic strains neutralized reactive oxygen types [32]. In a study by Michalickova et al., *Lactobacillus helveticus* was given to elite athletes (cyclists and endurance athletes) for 3 months and oxidative markers were evaluated. These authors showed that this probiotic had an antioxidant effect and reduced oxidative stress [33]. Similarly, in a study by Lamprecht et al., 23 athletes were given multi-strain probiotics (*Enterococcus faecium, Lactobacillus,* and *Bifidobacterium* strains) and intense exercise was applied for 14 weeks. The probiotics reduced intestinal permeability and positively affected exercise-induced protein oxidation [34]. In a state of chronic exercise metabolism, the maximal metabolic rate can reach 20 times higher than normal [35]. However, the metabolizing properties of probiotics also differ [36]. We interpret our findings to imply that the administration of the probiotic preparation utilized positively affected the exercise, thus, leading to the superior effects seen in the probiotics + exercise group.

## 5. Conclusions

In the present study, the effects of combining exercise and multi-strain probiotics were investigated, for the first time, on thiol/disulfide homeostasis as an oxidative stress parameter. We showed that the probiotics alone did not affect oxidative stress, but when used together with exercise, they significantly supported antioxidant capacity. As stated above, the opinion that the use of probiotics has a positive effect on antioxidant capacity contradicts the findings obtained in the group using only probiotics in our study. Considering that the composition of the gut microbiota of specific probiotic strains can affect gut neuro-motor function in different ways, it is thought that different types of probiotics may cause different physiological and biological responses. Thus, we believe that the types of probiotics we used in our study have the potential to be effective in reducing the damage that occurs at the cellular level, and therefore, to be possibly used to support athletes in sports that require intense and endurance training.

## Figures and Tables

**Figure 1 nutrients-14-03555-f001:**
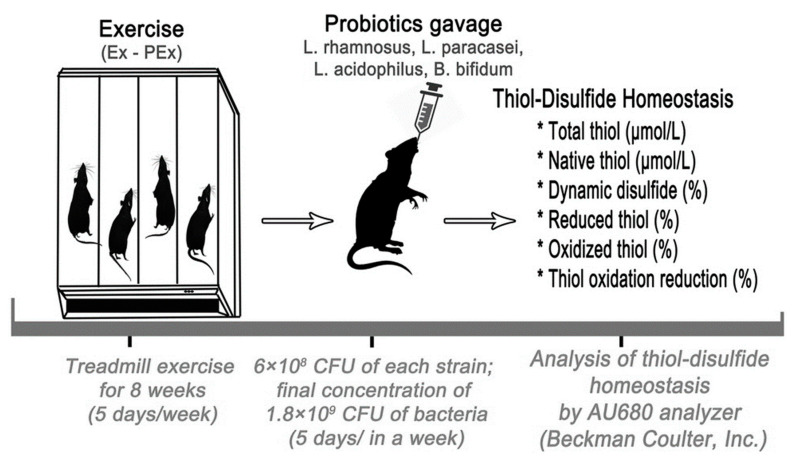
Experimental protocol of the study.

**Figure 2 nutrients-14-03555-f002:**
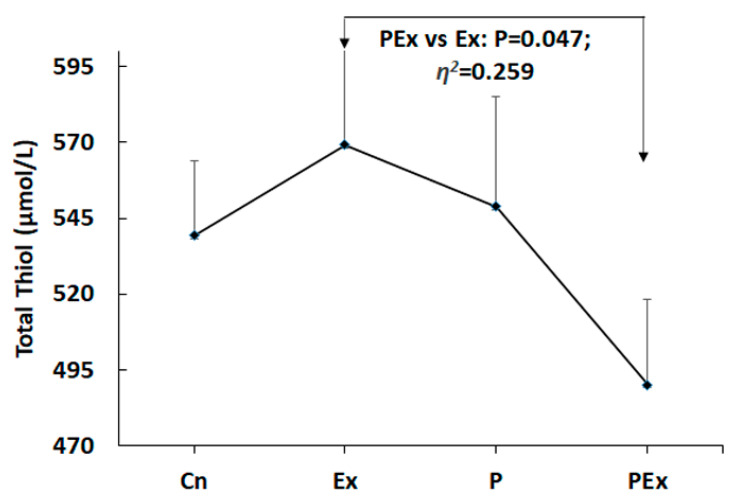
Total thiol levels. The total thiol level was found to be significantly higher in the EX group as compared with in the PEx group. The use of probiotics alone did not have any effect on the total thiol level.

**Figure 3 nutrients-14-03555-f003:**
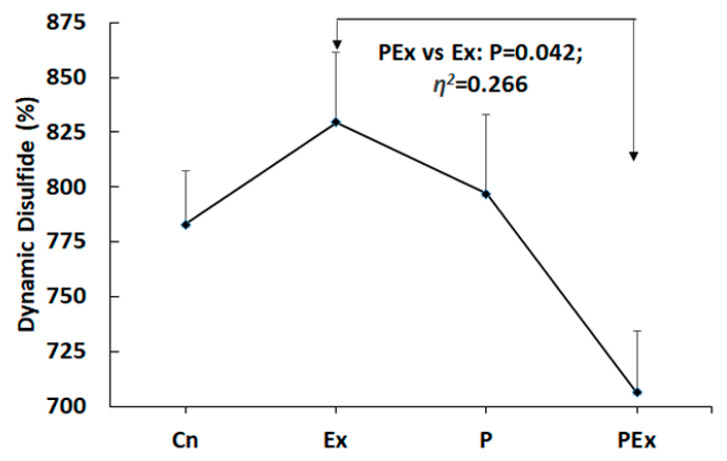
Dynamic disulfide levels. The dynamic disulfide level, which is considered to be an important indicator of oxidation, was found to be significantly lower in the PEx group as compared with in the Ex group. In addition, it was observed that the use of probiotics alone did not have any effect on cellular oxidation.

**Table 1 nutrients-14-03555-t001:** Findings of the thiol/disulfide parameters of the groups.

Parameters	Group	Mean	Std.D.	SEM	Eta Squared (η^2^)	P
**NT** **(μmol/L)**	Cn	147.86	13.171	4.978	0.228	NS
	Ex	154.57	9.947	3.760		
	P	150.43	17.587	6.647		
	PEx	137.00	10.893	4.117		
**TT** **(μmol/L)**	Cn	539.29	45.474	17.188	0.259	0.047
	Ex	**569.29**	52.261	19.753		
	P	548.86	65.162	24.629		
	PEx	**490.14**	47.291	17.874		
**DD** **(%)**	Cn	782.86	65.522	24.765	0.266	0.042
	Ex	**829.43**	85.094	32.162		
	P	796.86	96.020	36.292		
	PEx	**706.29**	74.220	28.052		
**RT** **(%)**	Cn	27.43	0.535	0.202	0.129	NS
	Ex	27.14	0.900	0.340		
	P	27.43	0.976	0.369		
	PEx	28.00	1.000	0.378		
**OT** **(%)**	Cn	145.14	1.215	0.459	0.187	NS
	Ex	145.71	1.890	0.714		
	P	145.00	1.291	0.488		
	PEx	143.86	1.574	0.595		
**TOR** **(%)**	Cn	18.86	0.690	0.261	0.074	NS
	Ex	18.86	0.690	0.261		
	P	18.86	0.690	0.261		
	PEx	19.29	0.756	0.286		

Eta Squared (η^2^) = small, 0.01; medium, 0.059; large, 0.138.

## Data Availability

The original contributions presented in the study are included in the article, further inquiries can be directed to the corresponding author.

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
