# Peer review of "The Use of Probiotics Combined with Exercise Affects Thiol/Disulfide Homeostasis, an Oxidative Stress Parameter"

_nutrients, 2022, doi:10.3390/nu14173555_

Round 1

Reviewer 1 Report

The introduction in the abstract is more like a goal than an introduction.

In the methodology there is information about the strain they gave every day, and in the abstract that every other day - there is a mistake somewhere.

My attention is - does this topic really fit the Nutrients journal? In my opinion, the study is not nutrition or nutrient related.

Reviewer 2 Report

In the paper "Probiotics use with exercise affects thiol-disulfide homeostasis,

an oxidative stress parameter," the authors attempted to describe the relationship between probiotics and exercise and their effects on oxidative stress. The oxidative stress indicator studied by the authors of the paper was thiol-disulfide.

I can say that the authors designed the scientific study correctly (according to the stated purpose of the study).

My doubts are raised in section 2.2 Please describe in more detail the course of probiotic administration procedures.

Please explain how the probiotic dosage was selected

I have no objection to the statistical methods used.

The authors were the first to show that a probiotic administered "with exercise" reduces dynamic-disulfide levels and significantly reduces oxidative damage. The authors conclude that the use of probiotics in high-intensity exercise sports can be used as an effective and practical tool to reduce oxidative stress.

The paper is nine pages of text (including bibliography). The course of the experiments and the results of the study are presented in 3 figures and 1 table. 

The authors cited 36 items of literature. In my opinion, a higher number of citations should have been included. In the medical literature we can find a very large number of interesting and valuable works on probiotics.

Reviewer 3 Report

The manuscript entitled "Probiotics use with exercise affects thiol-disulfide homeostasis, an oxidative stress parameter" provides valuable information and may be published in " Nutrients".

I have some suggestions for authors to improve further their study. A graphical abstract could support further the study.

Please add also the complete list of abbreviations.
